# Characterization of Four Medically Important Toxins from *Centruroides huichol* Scorpion Venom and Its Neutralization by a Single Recombinant Antibody Fragment

**DOI:** 10.3390/toxins14060369

**Published:** 2022-05-26

**Authors:** Hugo Valencia-Martínez, Timoteo Olamendi-Portugal, Rita Restano-Cassulini, Hugo Serrano-Posada, Fernando Zamudio, Lourival D. Possani, Lidia Riaño-Umbarila, Baltazar Becerril

**Affiliations:** 1Departamento de Medicina Molecular y Bioprocesos, Instituto de Biotecnología, Universidad Nacional Autónoma de México, Apartado Postal 510-3, Cuernavaca 62250, Mexico; hugo.valencia@ibt.unam.mx (H.V.-M.); timoteo.olamendi@ibt.unam.mx (T.O.-P.); rita.restano@ibt.unam.mx (R.R.-C.); fernando.zamudio@ibt.unam.mx (F.Z.); lourival.possani@ibt.unam.mx (L.D.P.); 2Investigador por México, CONACyT-Laboratorio de Biología Sintética, Estructural y Molecular, Laboratorio de Agrobiotecnología-Tecnoparque CLQ, Universidad de Colima, Carretera Los Limones-Loma de Juárez, Colima 28627, Mexico; hserrano0@ucol.mx; 3Investigadora por México, CONACyT-Instituto de Biotecnología, Universidad Nacional Autónoma de México, Apartado Postal 510-3, Cuernavaca 62250, Mexico

**Keywords:** affinity maturation, kinetic and thermodynamic parameters, phage display, scFv, scorpion toxins

## Abstract

*Centruroides huichol* scorpion venom is lethal to mammals. Analysis of the venom allowed the characterization of four lethal toxins named Chui2, Chui3, Chui4, and Chui5. scFv 10FG2 recognized well all toxins except Chui5 toxin, therefore a partial neutralization of the venom was observed. Thus, scFv 10FG2 was subjected to three processes of directed evolution and phage display against Chui5 toxin until obtaining scFv HV. Interaction kinetic constants of these scFvs with the toxins were determined by surface plasmon resonance (SPR) as well as thermodynamic parameters of scFv variants bound to Chui5. In silico models allowed to analyze the molecular interactions that favor the increase in affinity. In a rescue trial, scFv HV protected 100% of the mice injected with three lethal doses 50 (LD_50_) of venom. Moreover, in mix-type neutralization assays, a combination of scFvs HV and 10FG2 protected 100% of mice injected with 5 LD_50_ of venom with moderate signs of intoxication. The ability of scFv HV to neutralize different toxins is a significant achievement, considering the diversity of the species of Mexican venomous scorpions, so this scFv is a candidate to be part of a recombinant anti-venom against scorpion stings in Mexico.

## 1. Introduction

In Mexico, during 2020, the number of sting accidents in humans caused by scorpions was approximately 270,000 [1]. The most recent reports indicate that 14 species of scorpion of the genus *Centruroides* are possibly responsible for envenoming cases [2]. However, apart from the 14 species described, there are at least seven other species responsible for these intoxications [3]. Such is the case of *Centruorides huichol* scorpion located in Nayarit, Mexico, where, in 2020, 14,712 intoxications were reported [1]. This scorpion, previously classified within *C. noxius* species, was reclassified through a taxonomic analysis that allowed to name *C. huichol* as an independent species [4]. Due to the great diversity of scorpion species and the novel strategies to obtain specific antibodies directed against the toxic components, in the cases of species not yet studied, systematic work is required to identify and characterize the toxic components present in the venoms. The recovery of sufficient biological material to implement the generation of antibodies and/or fragments thereof, which are capable of neutralizing toxic components, is also important.

Various groups have ventured into using single-chain antibody variable fragments (scFv) as an alternative for generating anti-venoms against venomous animals [5,6,7]. This format has the advantage of being relatively small (27 kDa), which allows it to have better bioavailability and faster elimination [8]. On the other hand, its affinity can be improved relatively quickly (even to a higher level than natural immunization). Additionally, it is possible to extend the cross-reactivity through several cycles of directed evolution using site-directed [9] or random [10] mutagenesis. The characterization of its kinetic constants and thermodynamic interaction parameters can be evaluated efficiently through surface plasmon resonance [11].

From a phage-displayed repertoire of human scFvs and screened against Cn2 toxin (main toxin of *C. noxius* venom), scFvs 3F and C1 [12] were isolated, which recognize different epitopes of this toxin [13]. From scFv 3F, scFvs 6009F and 9004G were generated by directed evolution against Cn2 and Css2 (from *C. noxius* and *C. suffusus*, respectively). Through a combination of mutations, scFv LR was obtained, which contains V101F (HCDR3) mutation of scFv 6009F within the context of scFv 9004G, which is an efficient antibody capable of neutralizing the main toxins (Cn2 and Css2 respectively) and venoms of *C. noxius* and *C. suffussus* [10]. On the other hand, from the scFv C1, scFv 10FG2 was generated, by directed evolution, random and site-directed mutagenesis, and a combination of mutations [12,13,14,15,16]. The individual or joint use of these scFvs allowed to neutralize various scorpion venoms and toxins [10,16,17]. However, there are venoms such as *C. limpidus* whose neutralization by scFv 10FG2 is partial due to the presence of the Cl13 toxin, which is not recognized by 10FG2 [17]. Recently, the generation of scFv 11F of murine origin was published. This scFv recognized and neutralized Cl13 toxin and mixed with 10FG2, neutralized the whole venom of *C. limpidus* [18].

Identification of the toxic components for mammals of *C. huichol* venom, led to identify Chui5 toxin, whose sequence differs significantly from the one corresponding to toxins already recognized and neutralized by scFv 10FG2, explaining why it is not capable of efficiently neutralizing the whole venom. In this work, through the affinity maturation of 10FG2 toward Chui5 toxin by means of the construction of different libraries generated by random and/or site-directed mutagenesis, scFv HV was obtained. This scFv could neutralize Chui5 toxin and helped, in conjunction with scFv 10FG2 to optimally rescue intoxicated mice with the whole venom.

## 2. Results

*C. huichol* scorpion (Appendix A) was recently identified as a new species [4], and therefore a characterization process was initiated to confirm whether it is of public health importance, due to its potential toxicity to mammals. The toxicity of the venom in CD1 mice was evaluated and the LD_50_ of the fresh venom was determined using the “up & down” method [19] described in Materials and Methods section. This was 15.4 µg/20 g of mouse, which is in the intermediate range of toxicity with respect to the LD_50_ of other Mexican scorpion venoms [2].

### 2.1. Preliminary Neutralization Assays of C. huichol Venom with scFvs 10FG2 and LR

The scFvs generated in our group so far, have shown significant cross-neutralization capacity against various scorpion toxins. Based on these observations, mix-type assays were carried out [16], where 1 LD_50_ of fresh *C. huichol* venom was mixed with scFvs LR and 10FG2 at a molar ratio of 1:10:10 (venom: scFv LR: scFv 10FG2). To estimate the amount of scFv to be used, in this experiment it was assumed that the proportion of toxins corresponds to 10% of the venom (average value with respect to the venoms studied so far [16,18,20]). Control mice treated with 1 LD_50_ of venom, showed signs of severe intoxication 15 min after injection and three died within 2 h. The animals injected with a mixture of venom and the scFvs 10FG2 and LR showed a 5 h delay in the appearance of signs of intoxication, which were mild to moderate, and one animal died 12 h post-injection (Table 1). These results indicate that not all the medically important toxins present in the venom were efficiently neutralized, so it was necessary to identify the toxic components of this venom.

### 2.2. Characterization of Toxic Components

In order to know the characteristics of the toxic components of *C. huichol* venom, different chromatographic procedures were used. Venom extraction allowed obtaining 107 mg of soluble component, which was separated by molecular exclusion chromatography on a Sephadex G-50 column (see Materials and Methods details) obtaining 4 fractions (Figure 1a). To identify the toxic components, the toxicity of the fractions and sub-fractions that typically contain the toxins isolated in the different chromatographic steps was evaluated in mice. It is known that the FII fraction contains the peptides that are toxic to mammals [21,22,23]. This toxic fraction was subsequently processed by ion exchange chromatography on a carboxymethylcellulose (CMC) column, obtaining 14 sub-fractions (Figure 1b). Sub-fractions FII.9, FII.10 and FII.12 were toxic to mice at a dose of 30 µg of total protein/20 g mouse. These sub-fractions were subjected to reverse phase HPLC chromatography on a C18 column. Fraction FII.9 contains three main peaks, while fractions FII.10 and FII.12 only have one peak (Figure 1c). Each of these peaks were subjected to mass spectrometry to determine their purity and molecular masses. The FII.9-1 fraction (named Chui1), showed a molecular mass of 7405.8 Da, typical of toxins that affect voltage-gated sodium channels; however, mice injected with Chui1 did not show signs of intoxication. For this reason, Chui1 was not further analyzed and described here. Fractions FII.9-2, FII.9-3, FII.10-1, and FII.12-1, with masses in the same range, were lethal at a concentration of 1.7 µg/20 g mouse (Table 2). It was also possible to estimate their proportion within the venom. Toxins present in fractions FII.10-1 (Chui4 with 6.2%) and FII.12-1 (Chui5 with 5.7%), were the most abundant while the least abundant were FII.9-3 (Chui3) and FII.9-2 (Chui2) with abundances of 1.8% and 0.3% respectively.

To know the primary structure of the toxins, the pure peaks were processed for sequencing using the Edman degradation methodology [17]. The amino-terminal was directly sequenced to know the first 34, 35, 40, and 40 amino acids of the Chui2, Chui3, Chui4, and Chui5 toxins, respectively. Carboxyl-terminal sequence was determined by digesting each toxin with Asp-N and V8 proteases. The generated peptides, once purified, were sequenced. Identification of the cysteine residues (Figure 1d) was carried out by means of reduction and alkylation reactions [17].

To better understand the action of the venom, the activity of Chui3, Chui4, and Chui5 toxins was tested on human voltage gated sodium ion channels (hNav) of type: hNav1.1, hNav1.2, hNav1.3, hNav1.4, hNav1.5, hNav1.6, and hNav1.7. The three toxins act on hNav1.6 as β-type toxins, modifying the voltage dependence of the channels activation process. This means that, during toxins exposure, the hNav1.6 channels open at more negative membrane potential; meanwhile the maximal current is reduced (Figure 2). The three toxins were tested at a concentration of 200 nM and under these conditions Chui5 was the most active on the hNav1.6 channels, followed by Chui3 and finally Chui4. Chui5 toxin also had a weak β-effect on hNav1.5 channels, reducing the total current of these channels. Chui4 was also able to weakly shift the activation curves of the hNav1.2 and hNav1.4 (Appendix A). The values of the fitting parameters and of the current reduction are summarized in the Appendix A. Finally, in these conditions, it can be considered that the action of Chui3, Chui4, and Chui5 is mainly toward hNav1.6 channels. The amount of purified Chui2 was not enough for electro physiological assays.

### 2.3. Evaluation of the Recognition of scFvs LR and 10FG2 toward C. huichol Toxins

A mixture of scFvs LR and 10FG2 generated a delay in the time of the onset of signs of envenoming in mice treated with these scFvs, as well as a higher survival rate of the mice with respect to the control (Table 1). Due to these observations, the individual capacity of each scFv to interact with the different toxins of *C. huichol* was evaluated. Toxins were covalently immobilized on CM5 chips reaching bindings between 90 and 417 resonance units (RU). Molecular interactions with scFvs were determined with the BIAcoreX100. Figure 3 shows the sensorgrams of the interaction of the scFvs with the toxins. The evaluation of the scFvs was performed using a concentration of 100 nM for the scFv 10FG2 and 25 nM for the scFv LR. The recognition level of the scFv 10FG2 against Chui2, Chui3, and Chui4 toxins is relatively high while the interaction with Chui5 is much lower. In this last case, despite showing a good level of association it dissociates quickly (Figure 3).

In the analyses of scFv LR at 25 nM, it is observed that it also binds to three toxins to a good level (Chui2, Chui3, and Chui4); however, it dissociates much faster than scFv 10FG2. It is important to note for an antibody fragment to be neutralizing, it must show a minimal dissociation. Moreover, LR does not interact with Chui5 while scFv 10FG2 does. These results showed that scFv 10FG2 contributes to a higher extent in the neutralization of the toxins and for this reason, the kinetic constants of interaction were determined, as well as the evaluation of the neutralizing capacity was carried out. The determination of the affinity was achieved by evaluating the interaction of scFv 10FG2 by SPR at different concentrations (between 10 and 225 nM) with the four toxins at a temperature of 25 °C (Table 3). The K_D_ of scFv 10FG2 for Chui2 and Chui3 toxins are in the nM range and for Chui4 in the sub-nM range. From the k_off_, it is possible to calculate the residence time (T_R_) or the interaction time between the scFv and the toxins. The results showed that they are higher than previously determined for the scFv 10FG2 with other toxins, which were only 16 min [16]. The level of interaction of 10FG2 with Chui5 toxin is low, since it showed a K_D_ value of 12.2 nM and a T_R_ of less than 5 min.

Subsequently, neutralization assays of the Chui3, Chui4, and Chui5 toxins were performed with scFv 10FG2. Chui2 toxin was not evaluated because its proportion in the venom was very low (0.3%), thus we assumed that it would have a limited role in envenoming. Furthermore, its yield was insufficient to implement toxicity assays. For the other toxins, 1 LD_100_ was assayed with scFv 10FG2 in a 1:10 molar ratio (toxin: scFv). While controls showed 100% mortality with all toxins, scFv 10FG2 was able to completely neutralize the toxic effect of Chui3 and Chui4 since no signs of envenoming were observed and all individuals survived (Table 4). In view of the K_D_ and T_R_ of the scFv toward these two toxins and the Chui2 toxin (Table 3), it is very likely that the latter is also neutralized by 10FG2. In contrast, scFv 10FG2 with Chui5 toxin only managed to neutralize 33.3% of the mice, with severe signs of intoxication being observed (Table 4). Taking into account that Chui5 corresponds to 5.7% of the venom, the incomplete neutralization of *C. huichol* venom by scFv 10FG2 is explained.

### 2.4. Affinity Maturation of scFv 10FG2 against Chui5 Toxin 

scFv 10FG2 turned out to be the best candidate to be matured since it recognizes and partially neutralized Chui5 toxin. Based on these characteristics, it was decided to improve its interaction properties until it became a neutralizing scFv. For the first maturation process, a library constructed by site-directed mutagenesis at positions Y53 and S57 of HCDR2, Y59 of HFR3 and D102, as well as L105 of HCDR3, with a size of 3 × 10^7^ transformants was used. These positions were chosen from the analysis of the interaction of the scFv 10FG2 with Cl13 toxin from an in silico model (see Materials and Methods). Three rounds of screening against Chui5 toxin were performed, and from the third round, 180 random colonies were evaluated by ELISA assays. From this process, variant 9a was selected, which was the one that showed the highest signal in an ELISA assay against the toxin (Appendix A). The sequence of this variant beared the L105Q change in HCDR3, identifying position 105 as important for the interaction with the toxin. When determining the kinetic parameters of the interaction, a decrease in the value of the dissociation rate (k_off_) was observed, resulting in an improvement in both the affinity constant (K_D_; 8.2 nM) and the T_R_ (14 min; Figure 4a; Table 5). With this improvement, it was decided to determine the LD_50_ of Chui5 toxin and the neutralizing capacity of the 9a variant. Again, the “up & down” method was used, as in ref [24], and it was determined that the LD_50_ of Chui5 toxin was 1.1 μg/20 g of CD1 mice. For the neutralization assay, 2 LD_50_ of the toxin were used with a molar ratio of scFv 1:10. scFv 9a increased the extent of protection to 60% of the mice and with an average delay time of the onset of signs around 99 min, while scFv 10FG2 protected 16.7% of mice and the onset of signs at around 18 min after injection (Table 6).

For the following maturation process, clone 9a was used as a template from which a library was generated by random mutagenesis [14] with a size of 2.8 × 10^7^ transformants. After the third round of screening, 180 random clones were evaluated and the B15a variant was selected which maintained L105Q change and incorporated K65Q mutation in HFR3. The protein was expressed and evaluated for determining its kinetics interaction with Chui5. The parameter that changed was the dissociation rate whose value is lower resulting in a better interaction with a K_D_ = 7.3 nM (Figure 4a, Table 5). This result represents a slight increase in T_R_ (17 min). Its ability to neutralize 2 LD_50_ of Chui5 in mice using a 1:10 molar ratio (toxin: scFv) was also evaluated. The result showed that B15a variant, despite further delaying the onset of signs (around 125 min), only managed to neutralize 33.3% of the mice (Table 6).

From an analysis of cross-reactivity level that 10FG2 showed with other toxins from scorpions of the genus *Centruroides* (Figure 4b), it was determined that positions 8 and 9 in the toxins are important for the interaction with amino acids K65 and L105 of the scFv 10FG2 [14,15,16] which are different in Chiu5 toxin. For this reason, it was decided to mutagenize positions 65 and 105 to saturation, including random mutagenesis using variant 9a as template. Of the three rounds of screening carried out, the second and third ones were done under more stringent conditions incubating the phage-antibodies bound to toxin absorbed in the immunotube in the presence of 0.5 M and 1 M guanidinium chloride solution in PBS, respectively, for 30 min at 37 °C (See Materials and Methods). Likewise, from the third round, 528 clones were evaluated, of which the best 18 clones were subjected to sequencing. The sequences were aligned, mainly observing variants with changes at positions 65 and 105. In addition, other mutations (of lower recurrence) were detected at other positions, product of random mutagenesis (Appendix A). The variant called HV was the one that showed the highest signal in the ELISA assay and did not present amber-type stop codons. This scFv, relative to B15a, kept K65Q change and acquired three changes: F29Y in HCDR1, L105G in HCDR3, and G154E in LFR1. HV variant was sub-cloned into the pSyn1 vector and expressed. With the purified protein, the kinetic parameters of interaction with the toxin were determined. At 25 °C, HV variant shows minimal k_on_ enhancement over B15a variant; however, k_off_ shows a slight improvement resulting in a K_D_ of 6.9 nM and a T_R_ of 15.5 min (Figure 4a, Table 5). Neutralization of 2 LD_50_ of Chui5 with HV at a 1:10 molar ratio (toxin: scFv) protected 75% of mice and showed a delay in the onset of symptoms around 138 min (Table 6). This result is significantly better than that obtained with variants 9a and B15a, turning scFv HV into a candidate for further characterization. To achieve a better neutralization of Chui5 toxin, the amount of scFv HV was increased, 1 LD_50_ was assayed in a molar ratio of 1:20 reaching 100% protection of mice and the absence of envenoming signs (Table 6).

### 2.5. Determination of the Kinetic Constants of scFv HV against C. huichol Toxins

Due to the neutralizing properties of the scFv HV against Chui5 toxin, it was decided to evaluate the binding levels with Chui2, Chui3, and Chui4 toxins (Figure 3). It was observed that the changes present in this antibody fragment modified its interaction toward these toxins. When comparing the kinetic constants with those of scFv 10FG2, it is observed that the K_D_ are maintained in the same order (Table 3). For the case of Chui2 toxin, scFv 10FG2 showed a T_R_ of 60.4 min whereas for scFv HV it was 34.2 min. Chui3 toxin showed a small change since the T_R_ was 53.1 min with scFv 10FG2, but for scFv HV it was 48.8 min. The most affected kinetic constants were with Chui4, since T_R_ values of 241.5 and 104.8 min were obtained with 10FG2 and HV, respectively. Neutralization of Chui5 toxin with scFv HV was 100%, with a T_R_ of 15 min. Therefore, it could be assumed that scFv HV also neutralized Chui2, Chui3, and Chui4 toxins because scFv HV residence times with these toxins are higher. This cross-neutralization has also been observed with other members of this family [16].

### 2.6. Neutralization of C. huichol Venom with scFv HV

Considering the cross-reactivity of scFv HV toward Chui2, Chui3, Chui4, and Chui5 toxins and their K_D_ and T_R_ values against them, a fresh venom neutralization assay was performed with this variant both in mix-type (venom: scFv), as well as rescue assays (see Materials and Methods). With regard to mix-type assay, the ability of scFv HV to neutralize 1 LD_50_ of venom in a 1:10 molar ratio (venom: scFv) was evaluated, as well as in combination with scFv 10FG2 in a molar ratio 1:10:10 (venom: scFv 10FG2: scFv HV). A 100% protection was achieved in both modalities, with the difference being the presence of mild signs of intoxication in those mice that only received scFv HV and the complete absence of symptoms in those who received the mixture of the two scFvs. Because of these results, in the following assay the amount of scFv HV was maintained while the number of LD_50_ was increased. In this way, when neutralizing 2 LD_50_, the molar ratio was 1:5 (venom: scFv HV) where the neutralization of 66.7% of the mice was reached with the appearance of mild signs of intoxication in those that survived. Subsequently, 3 LD_50_ with a 1:3.3 molar ratio (venom: scFv) were assayed, reaching a neutralization percentage of 33.3% of the mice, which presented moderate signs of intoxication. These results showed that molar ratios of at least 1:10 (venom: scFv) are required to achieve neutralization with scFv HV. Based on these results and the ones corresponding to the neutralization of the scFv 10FG2 with the other toxins, the capacity of neutralizing 5 LD_50_ of venom using the combination of scFvs 10FG2 and HV in a 1:5:5 molar ratio, 100% of the mice were protected, although with the presence of moderate signs of intoxication (Table 7).

On the other hand, scFv HV was evaluated in a rescue assay, which more closely mimics the treatment of a scorpion sting envenomation. The experimental group of mice was injected with 3 LD_50_ of fresh venom and 10 min post-injection the scFv HV was applied at a 1:10 molar ratio. One hundred % of the envenomed mice were rescued. Severe signs of intoxication disappeared around 2 h after the administration of the scFv HV, and the total disappearance of the signs occurred around 48 h later (Table 7).

### 2.7. Thermodynamic and Structural Analysis of the Improvement of scFv HV

scFv HV was able to neutralize Chui5 toxin, so the binding characteristics of scFvs were analyzed through thermodynamic parameters of interaction and by in silico models of the Chui5-10FG2 and Chui5-HV complexes.

#### 2.7.1. Thermodynamic Analysis

The values of the interaction kinetic constants of the scFvs HV and B15a at 25 °C with Chui5 toxin are very similar (Table 5); however, the differences in terms of neutralizing capacity were clear, with HV variant being the best (Table 6). To understand this property, the kinetic constants of interaction were determined at different temperatures and, with the K_D_ from these results, it is possible to calculate thermodynamic parameters through the Van’t Hoff expression [11,25,26] (See Materials and methods). For each interaction of Chui5 with scFvs, a Van’t Hoff plot was created (Appendix A). The slope of the straight line in the Van’t Hoff plot determines the stability of the scFv binding with Chui5 toxin. The absolute value of the slope of scFvs 10FG2 and B15a is very similar (1316 and 1299 respectively) but the one for scFv HV was the lowest (814.6) suggesting that Chui5-HV complex establishes a more stable interaction than the Chui5-B15a complex when the temperature varies. When calculating the thermodynamic parameters of the binding, we observed in the three complexes that ΔG is negative, indicating that these bindings are spontaneous (Table 8). The negative variation of ΔH indicates the formation of weak interactions such as hydrogen bonds, salt bridges, etc., [26]. Both scFv 10FG2 and B15a showed equal ΔH values (−2.6 kcal·mol^−1^) that are more negative than those calculated for scFv HV (−1.6 kcal·mol^−1^). These parameters suggest that scFv HV could be missing interactions with Chui5 toxin. The positive variation of the ΔS indicates the degree of freedom of movement between the scFv and the toxin [26]. Chui5-HV interaction is the one with the highest ΔS (31.9 cal·mol^−1^·K^−1^) compared to Chui5-10FG2 and Chui5-B15a interactions (27.3 and 28.6 cal·mol^−1^·K^−1^ respectively), suggesting a greater freedom of movement in the complex Chui5-HV (Table 8).

#### 2.7.2. In Silico Models of Chui5 Toxin with scFvs 10FG2 and HV

To understand how the molecular interactions of scFv HV with Chui5 toxin were improved, two in silico models of the binary Chui5-10FG2 and Chui5-HV complexes were generated by in silico mutagenesis and modeling using the crystallographic structure of scFv RU1-Cn2-LR scFv ternary complex (PDB entry 4V1D) [13]. The analysis focused on the mutated positions of scFv HV. Hence, the cation-π interaction of 3.3 Å between F29 of HCDR1 and R72 in the scFv 10FG2 is conservatively changed to the interaction with Y29 in scFv HV. Nevertheless, Y29 residue also forms a hydrogen bond of 2.8 Å with the main chain of T78 (Figure 5a). In contrast, while G154 of LFR1 has no interactions in 10FG2 variant, the change to E154 in HV generates a hydrogen bond of 3.1 Å with the side chain of Q219 (Figure 5b). These two new hydrogen bonds may contribute to the intra-chain stability of scFv HV. Regarding the changes in the toxin-scFv interaction, while K65 of HFR3 forms a strong salt bridge of 3.0 Å with D62 in scFv 10FG2, the change to Q65 in scFv HV generates a weak hydrogen bond of 3.8 Å (Figure 5c,d). Moreover, although the side chain of L105 in scFv 10FG2 seems to favor fitting the HCDR3 region with the toxin, this adjustment shows a steric hindrance between the bulky residues Y8/Y59 of Chui5-10FG2 complex, respectively (Figure 5e). Accordingly, this observation suggests that the low affinity of scFv 10FG2 toward Chui5 (Table 5) could be related to this weak surface complementarity between the scFv and the toxin. Remarkably, L105G change in HV removed the steric hindrance between Y8/Y59 of Chui5-HV complex (Figure 5f), generating a flexible zone that favored a proper fitting of the HCDR3 region with the toxin.

## 3. Discussion

In recent years, progress has been made in the study of the diversity of Mexican scorpion species of medical importance [2,3]. One of them recently described is the scorpion *C. huichol* [4]. The present work showed that the venom of this species is lethal to mammals. The toxicity is due to the presence of four lethal toxins in this venom: Chui2, Chui3, Chui4, and Chui5. It is noteworthy that Chui3 sequence is the same as the one of Cn3 toxin from the venom of *C. noxius* [27]. Of the three more abundant toxins, it was confirmed that they showed activity on hNav1.6 human sodium channel. In the case of Chui5 toxin, it also modified the activity of hNav1.5 channel. The presence of four components with a total abundance of 14% is significant since, typically, the abundances of the toxins correspond to less than 10% in the venom of Mexican scorpions [16,18,20].

*C. huichol* venom was partially neutralized by scFv 10FG2, which is capable of neutralizing various venoms and toxins from scorpions of the genus *Centruroides* [16]. This partial neutralization is due to the presence of Chui5 toxin with an abundance of 5.7% and whose primary structure varies considerably from those already neutralized by this scFv. In this work, Chui5 toxin was used to perform three in vitro affinity maturation processes of scFv 10FG2. In the first process, variant 9a with L105Q mutation was obtained. The change in this position, compared to the others obtained, was the most important in terms of improvement of K_D_ and T_R_. This variant delayed the onset of intoxication signs, as well as their intensity due to a considerable increase in T_R_; however, the toxic effect of Chui5 was not neutralized. Position 105 of the parental scFv C1 was previously changed in processes of directed evolution against toxin Cll1 from *C. limpidus* venom. In that work, the scFv 202F (with various mutations including M105L) was isolated, whose kinetic and neutralizing characteristics were improved when compared to its parent (scFv C1) [14]. In addition, position 105 has been shown to be an important point of hydrogen bond-type interactions between several amino acids of various toxins (positions 5, 6, 8, 33, 55, and 56) and scFv 10FG2 [16]. Variant 9a was used as template for the second maturation process, obtaining variant B15a that maintains L105Q mutation and incorporates K65Q mutation. This variant showed an improved K_D_ value compared to 9a and a higher T_R_, delaying the onset of signs; however, its neutralizing capacity was lower. Previously, a model of the interaction between toxin Cn2 and scFv RJI-2 was generated which showed that position K65 of the scFv forms a hydrogen bond with N9 of the toxin. This observation prompted to identify RJI-2 as a target for mutagenesis by applying a rational design strategy. In this way, ER-5 variant was generated with K65R mutation, which neutralizes toxins Cn2, Cll1, and Css2, as well as the venom of *C. suffussus* [15]. Initially in this work, L105Q and K65Q changes were shown to be relevant for the affinity maturation toward Chui5 toxin. However, the results of the neutralization and the interaction kinetic constants indicated that the selected changes are not necessarily the optimal ones to obtain a neutralizing scFv. With this rational, positions 65 and 105 of scFv 9a were again mutagenized to saturation and random mutagenesis were performed at the same time. From this process 18 clones were isolated and sequenced, letting to analyze the frequency of occurrence of the amino acids at positions 65 and 105 (Appendix A). It was observed that 50% of the sequences showed a glutamine at position 65 and 50% a glycine at position 105. Only 4 clones of the 18 evaluated carried changes K65Q and L105G, which are also present in the best variant, scFv HV. This scFv showed the highest recognition signal in ELISA assays (data not shown) and the absence of amber stop codons (*E. coli* TG1 replaces amber stop codon by Q). This variant with an improved K_D_ despite a decreased T_R_ value with respect to scFv B15a, was able to neutralize Chui5 toxin. The evaluation of scFv HV with the other toxins of the *C. huichol* venom showed that the mutations present decreased affinity and T_R_ (Table 3). However, these K_D_ and T_R_ values are similar to those previously reported for neutralizing members of this family of scFvs [16,17].

Due to the affinity properties, the cross-reactivity and the ability of the scFv HV to neutralize Chui5 toxin, its neutralization capacity against *C. huichol* venom was evaluated. In mix-type assays, the neutralization of 1 LD_50_ with scFv HV and with the combination of scFvs 10FG2 and HV was evaluated using a 1:10:10 molar ratio (venom: scFv(s)). In both groups, the neutralization was 100%. The number of LD_50_ was also increased, decreasing the venom: scFv molar ratio and it was observed that the neutralization decreased proportionally to the increase in LD_50_ until a partial neutralization of 33% was obtained with 3 LD_50_. These results lead to consider the use scFv 10FG2 as a neutralizing complement since it neutralizes other toxins except Chui5 (Table 7). At molar ratios 1:10:10, neutralization occurred without signs of intoxication; thus, it was tested at lower molar ratios. The assay was made with 5 LD_50_ of venom mixed with scFvs 10FG2 and HV at molar ratio 1:5:5 (venom: scFv 10FG2: scFv HV). A 100% neutralization was reached but with moderate signs of envenoming as expected due to this extreme challenge. Rescue assays were also carried out since they are closer to what happens during scorpion sting accidents. Only scFv HV was used in the rescue assays. Mice were envenomed with 3 LD_50_ and 10 min later injected with the scFv; all mice survived the challenge. The disappearance of the strong signs of intoxication (sialorrhea, respiratory stress) was evident at around 2 h post injection, although discomfort in the area of the application of the venom disappeared around 48 h later. If we consider the mix-type neutralization assay of 5 LD_50_ with scFvs 10FG2 and HV, the presence of 10FG2 in rescue assays can complement the neutralization of the venom and turn it even more efficient.

When analyzing the mutations of scFv HV, it was found that both the mutations in the FRs and those of the CDRs of the heavy chain variable domain contribute to the improvement of the neutralizing properties. Mutations F29Y and G154E seem to contribute to the stabilization of scFv HV at higher temperatures, which allowed the maintenance of binding to the toxin observed in the Van’t Hoff plot. It is likely that it is the result of the formation of new hydrogen bonds between Y29 and E154 with other residues of the scFv. Possibly, these mutations were favored by the use of guanidinium chloride in the screening rounds by eliminating unstable variants [28] and/or weak interactions. The differences observed in the values of the slopes in the Van’t Hoff plot correlate with the better neutralization efficiency of scFv HV compared to that of scFv B15a that do not possess these changes. Increased stability has also been observed in scFv LR [10] and LER [29] scFvs with mutations in the FR region for example. Structural analysis of changes K65Q and L105G in the interaction area, suggests the conformation of a more flexible contact area, without steric hindrance for the accommodation of bulky amino acids such as Y8 of Chui5 toxin and Y59 of scFv HV. This rearrangement of the interaction forces, in terms of thermodynamic parameters, would be favoring binding. The negative ΔG of all interacting complexes indicates that the binding would be spontaneous and would form a stable complex. The ΔH of the Chui5-HV complex is slightly less negative than Chui5-B15a and Chui5-10FG2 complexes, suggesting that original interactions of L105 were possibly lost [16], correlating with the decrease of the interaction of scFv HV with Chui2, Chui3, and Chui4 (Figure 3 and Table 3). This loss of interactions is reflected in the increase in ΔS in the Chui5-HV complex, which suggests a greater degree of movement of amino acid residues in the interaction zone of the toxin and scFv [30].

Cross-reactivity with toxins from the venom of different species is a highly desirable phenomenon in those scFvs that would be selected to form part of an anti-venom. In the case of scFv HV, it was observed that it recognized a recently identified *C. tecomanus* toxin too (unpublished data). We have successfully expanded the cross-neutralization of scFv 10FG2, so it would be very useful to have the structure of the Chui5-HV complex to optimize this interaction and make the neutralization of these and other toxins even more efficient.

## 4. Conclusions

The characterization of the venom of *C. huichol*, the processes of directed evolution, the characterization of the scFv HV in terms of cross-reactivity, neutralizing capacity of the venom of *C. huichol*, and the analysis of kinetic and thermodynamic interactions, have allowed to propose this scFv to be part of a novel recombinant anti-venom against the sting of Mexican scorpions.

## 5. Materials and Methods

### 5.1. Venom and Reagents

*C. huichol* scorpions were purchased from the company Octolab (Xalapa, Veracruz, Mexico). The venom was obtained by electro stimulation as reported [21]. The milked venom was dissolved in tetra-distilled water and centrifuged at 25,000× *g* rpm at 8 °C for 20 min. Soluble protein was quantified by spectrophotometry at λ = 280 nm. It was assumed that 1 AU = 1 mg/mL. All reagents used were of analytical grade.

### 5.2. Enzymes

Taq polymerase, T4 DNA ligase, and restriction enzymes Not1 and Sfi1 were purchased from Thermo Fisher (Waltham, MA, USA). V8 and AspN enzymes were purchased from Roche (Basel, Switzerland).

### 5.3. Purification, Analysis by Mass Spectrometry, and Sequencing of Toxins

Purification, analysis of pure toxins by mass spectrometry, and Edman degradation sequencing were performed as reported [17]. With the sequence, the theoretical molar extinction coefficient was obtained: 23,420 M^−1^·cm^−1^ for Chui2, 21,930 M^−1^·cm^−1^ for Chui3, and Chui4 and 24,910 M^−1^·cm^−1^ for Chui5. These data were used to calculate the concentration of the toxins with the formula A = εbc where A is the absorbance, ε is the molar extinction coefficient (M^−1^·cm^−1^), b is the width of the cell (cm), and c is the concentration (M).

### 5.4. Alkylation and Enzymatic Digestion of Toxins

Alkylation and digestion of the toxins were carried out as reported [17] with some modifications. To digest Chui2 toxin, 0.5 μg of the AspN protease was used (specific cut at the amino side of the Asp residue) and to digest Chui3, Chui4, and Chui5 toxins, 2 μg of the V8 protease were used (specific cut at the carboxyl side of the Asp and Glu residues).

### 5.5. Electro Physiology Assays

The experimental strategies used for the culture of HEK cells expressing hNav 1.1–hNav 1.6 sodium channels and CHO cells expressing hNav 1.7 sodium channels, as well as the conditions for the electrophysiological characterization of Chui toxins were previously described [17,24]. Data are the mean of 3–7 cells ± standard error and a Paired Sample t Assay was applied to calculate the significance at 0.05 level of the difference between control and toxin conditions.

### 5.6. scFv Expression

The DNA segments containing the genes that encode the selected scFvs, were subcloned into the pSyn1 (AmpR) vector as reported [12]. Briefly, electro competent *Escherichia coli* TG1 [F’*tra*D36 *proAB lacI_q_ZΔM15*] bacteria were electroporated with the recombinant pSyn1 constructs and grown in SOC medium for 1 h to be subsequently cultured in YT2X solid medium with 200 μg/mL ampicillin and 1% glucose. A pre-inoculum was generated from an individual colony in YT2X/Amp/Glu medium, incubated overnight (ON) and subsequently inoculated at a 1:100 dilution in YT2x/Amp medium. The culture was incubated up to an optical density (at λ = 600 nm) of 1.1 to be induced with 1 mM IPTG for 6 h at 37 °C. The bacterial culture was centrifuged, the supernatant discarded, and the pellet was resuspended with PPB buffer and 5 mM MgSO_4_. Both supernatants were recovered, mixed, and dialyzed two times against PBS 1X with agitation at 4 °C. The dialysate was passed through a Ni^2+^-NTA column previously equilibrated with 20 mM imidazole. The column with the attached scFvs was washed with 35 mM imidazole and subsequently eluted with 350 mM imidazole. Finally, the eluted samples were subjected to size exclusion chromatography using a Superdex TM 75 10/300 G L column (GE Healthcare) recovering the scFvs in PBS 1X. The molecular weight and molar extinction coefficient of each scFv was calculated theoretically using its sequence. Protein concentration was calculated by UV spectrophotometry λ = 280 nm with their molar extinction coefficient which were 49,765 M^−1^·cm^−1^ for three scFvs 10FG2, 9a and B15a and 51,255 M^−1^·cm^−1^ for HV.

### 5.7. Directed Evolution and Screening

The affinity maturation process included the generation of phage-displayed antibody libraries subjected to site-directed and/or random mutagenesis. For all the generated repertoires: the amplification products obtained were purified by means of a preparative 0.8% agarose gel with 1X TAE buffer. Analytical gels were made with 1% agarose in 1X TBE buffer. The PCR conditions were the same as earlier reported [12]. Bacterial cultures from the libraries were cryo-preserved with 20% glycerol and stored at −35 °C until use. Three rounds of screening were carried out as reported [12] with some modifications that are described below. Bacterial colonies were randomly selected from the 3rd round of screening and evaluated by ELISA of soluble scFvs. Those clones with a higher signal than their respective control were prepared for sequencing with the Dir [12] primer at Unidad de Síntesis y Secuenciación de DNA of the IBt-UNAM.

#### 5.7.1. First Library with Site-Directed Mutagenesis

A mutagenic library of the scFv 10FG2 with saturation mutations at positions 53, 57, 59, 102, and 105 was constructed. In separate PCR reactions using 10FG2 as template and two primer pairs Dir [12]-CDR2REVC1 (5′-C TGC ATA KYW TTT SNN ACC TCC ACC MTM TGA TAT AAC-3′) and CDR3DIRC1 (5′-GCC CGC NNS TGC CTA NNS TGC AGC GAC-3′)-cMyc[12], two mega-primers were generated as reported [12] containing saturating mutations at positions 53, 57, and 59 as well as 102 and 105 respectively. To obtain the complete gene, PCR products were used as mega-primers in a second PCR step with the same template (10FG2). The complete gene was digested with the enzymes Not1 and Sfi1 and ligated with T4 ligase into the phagemid vector pSyn2 previously digested with the same restriction enzymes. The ligation product was transformed into electro-competent *E. coli* TG1 cells and resuspended in SOC medium. After 1 hour of incubation at 37 °C, an aliquot was taken for the determination of the repertoire titer; the rest was resuspended in 50 mL of YT2x/amp/glu medium and grown ON at 150 rpm at 30 °C for cryo-preservation. The amount of Chui5 used for biopanning was 4, 1, and 0.2 μg, and the PBS/PBS-Tween20 washes were 10/10, 10/10, and 10/20 for the first, second, and third rounds, respectively.

#### 5.7.2. Library Construction with Random Mutagenesis

Clone 9a was used to generate the library with random mutagenesis by error-prone PCR as reported [14]. The amount of Chui5 used for biopanning was 4, 2, and 2 μg, and the PBS/PBS-Tween20 washes were 10/10, 12/15, and 15/15 for the first, second, and third rounds, respectively. The phage-antibodies from the 3rd round had an additional wash with 100 mM NaCl.

#### 5.7.3. Construction of the Second Library with Site Directed Mutagenesis

In separate PCR reactions, with template 9a and two primer pairs Dir [12]-Q65Xrev (5′-GAA TCG GCC SNN CAC GGA GT-3′) and Q105XDir (5′-GC CTA NNS TGC AGC GAC TGG-3′)-cMyc [12], two mega-primers were generated as reported [12] containing saturation mutations at positions 65 and 105 respectively. To obtain the complete gene, the 204 bp and 512 bp PCR products were used as mega-primers in a second PCR with the same template (9a). Furthermore, this second PCR was done under the same conditions to generate a library with random mutagenesis [14]. For the library construction, the complete gene was processed as mentioned before. The amounts of Chui5 toxin used for screening were 2, 1.5, and 1 μg for the first, second, and third rounds, respectively. PBS/PBS-Tween20 washes were 13/13 for the three rounds. For the 2nd and 3rd round, the phage-antibodies bound to the immunotube and prior to washing with PBS/PBS-Tween20, were incubated with a 0.5 M and 1 M guanidinium chloride solution in PBS, respectively, for 30 min at 37 °C.

### 5.8. Surface Plasmon Resonance Measurements

The binding kinetic constants of the scFvs 10FG2, 9a, B15a, and HV with the Chui2, Chui3, Chui4, and Chui5 toxins were determined using the BIAcoreX100 (GE, Uppsala, Sweden) biosensor system. Total of 500 ng of the different toxins were dissolved in 200 μL of MES buffer [10 mM of 2-(N-morpholino)ethanesulfonic acid pH6]. Twenty microliter of the mixture with each toxin was attached to cell 2 of a CM5 chip using the amine coupling kit with a flow rate of 5 μL/min. The binding RU were 90, 417, 367, and 170 for Chui2, Chui3, Chui4, and Chui5 toxins, respectively. Cell 1 (no toxin bound) was used as a reference. One hundred microliter of serial dilutions (10–225 nM) of each scFv were injected on each of the chips with a flow rate of 30 μL/min. The equipment was set at 25 °C. The dissociation time was 800 s. To regenerate the chips, 10 or 20 mM HCl was used as required. The sensorgrams used for the determination of the kinetic constants were corrected by subtracting the signal from the reference cell and injecting buffer/blank in the cases where it was necessary. The constants were determined with the BIA-evaluation version 3.1 program using the Langmuir mathematical model (1:1); the maximum Chi^2^ value allowed for all assays was 3. All determinations were performed in duplicate.

### 5.9. Toxin Neutralization Assays

All in vivo neutralization assays were approved by the Bioethics Committee of the Instituto de Biotecnología of the Universidad Nacional Autónoma de México IBt-UNAM (Project number: 413). The LD_50_ of the Chui5 toxin were determined as reported [24]. Neutralization assays of scFv 10FG2 were carried out with Chui3, Chui4, and Chui5 toxins. For the control groups, 6 female CD1 mice of 18–20 g were used and they were injected with a lethal amount of each toxin (Table 4). For the experimental group, mixtures of the toxins were made with the scFv 10FG2 at a molar ratio of 1:10 (toxin: scFv). For Chui5 toxin neutralization assays with scFvs 9a, B15a, and HV, the control and experimental groups ranged from 4 to 6 mice. Control groups were injected with 2 LD_50_ (Table 6). For the experimental groups, scFvs were mixed with Chui5 toxin in a 1:10 molar ratio. Additionally, a pool was made with a mixture of 1 LD_50_ of Chui5 with the scFv HV at a 1:20 molar ratio. All mixtures were incubated for 30 min at room temperature (~25 °C) and injected intraperitoneally in a volume of 100 µL. Surviving animals were observed for 48 h. In order to avoid unnecessary suffering, the animals that showed irreversible envenoming effects, were sacrificed according to the bioethical protocols approved by the IBt-UNAM.

### 5.10. Neutralization Assays with Whole Fresh Venom of C. huichol

Neutralization assays were performed with freshly collected scorpion venom to ensure the highest toxicity. The soluble part of the venom was quantified by spectrometry at λ = 280 nm and its LD_50_ was determined as reported [2].

#### 5.10.1. Classic Protection (Mix-Type Assays)

Female CD1 mice of 18–20 g weight were used for all assays. For the neutralization assay with scFvs 10FG2 and LR, a control group of 4 mice was used, which were administered with 1 LD_50_ of venom. The experimental group consisted of 4 mice injected with 1 LD_50_ and a mixture of scFvs 10FG2 and LR with a 1:10:10 molar ratio (venom: scFv 10FG2: scFv LR) assuming that the toxins represent 10% of the venom. For the neutralization assay with 10FG2/HV and HV alone, a control group of 6 mice injected with 2 LD_50_ was used. For experimental groups of the scFvs 10FG2/HV mix-type assays, 1 LD_50_ and 5 LD_50_ of venom were used with molar ratios of 1:10:10 and 1:5:5, respectively. For scFv HV alone groups 1 LD_50_, 2 LD_50_, and 3 LD_50_ of venom were used keeping the amount of scFv fixed to obtain different molar ratios of scFv (Table 7) assuming that the toxins represent 14% of the venom. All mixtures were incubated for 30 min at room temperature (~25 °C) and injected intraperitoneal with a volume of 100 µL.

#### 5.10.2. Rescue Assays

For the rescue assays, 12 CD1 female mice weighing 18–20 g were injected with 3 LD_50_ of fresh venom. About 10 min post-injection, 6 of such mice were administered with scFv HV in a 1:10 molar ratio assuming that the toxins represent 14% of the venom. Both injections were done intraperitoneal with a volume of 100 μL.

### 5.11. Calculation of Thermodynamic Parameters

The kinetic constants of scFvs 10FG2, B15a, and HV with Chui5 toxin were generated as mentioned before. The determinations were made at temperatures of 15, 20, 25, 30, and 35 °C. The Van’t Hoff equation in terms of K_D_ is represented mathematically as
(1)$$\Delta G={\mathrm{RTlnK}}_{D}=\Delta H-T\Delta S$$
where ΔG is the Gibbs free energy, R is the universal ideal gas constant (1.987 cal·K^−1^·mol^−1^), T is the temperature in Kelvin, K_D_ is the affinity constant, ΔH is the enthalpy, and ΔS the entropy. So, the equation becomes
(2)$${\mathrm{lnK}}_{D}=\frac{\;\Delta H}{R}\frac{1}{T}-\frac{\;\Delta S}{R}$$

This equation is a representation of the equation of a straight line of the form
(3)y=mx+b

To calculate the values of the thermodynamic parameters, a Van’t Hoff plot was made, which is obtained by plotting the inverse of the temperature (in Kelvin) on the X-axis against the natural logarithm of the K_D_ on the Y-axis. The values of the slope, obtained with the linear regression of 1/T vs lnK_D_, allow to calculate the ΔH and the y-intercept allows to calculate the ΔS; ΔG at 25 °C is calculated using Equation (1). Linear regression was performed with the GraphPad Prims 6 program for Windows version 6.01; the multiple correlation coefficient R^2^ = 0.95 was the minimum value allowed to assume that the graph is a straight line [30].

### 5.12. In Silico Mutagenesis and Modeling

The models of the binary complexes were made by in silico mutagenesis based on the crystallographic structure of the ternary complex scFv RU1-Cn2-LR scFv (PDB entry 4V1D [13]) eliminating the LR structure. We used this complex because the antibody fragments mentioned in this work come from the parental scFv C1, from which the scFv RU1 was derived. scFv RU1 gave rise to scFv 10FG2, and from this scFv HV was obtained. The identity between scFvs RU1 and HV is 96% (Appendix A). The models generated were scFvs 10FG2 and HV with Chui5 toxin. The Coot [31] standard rotamer library was used to generate the models, which were subsequently subjected to energy minimization using the YASARA [32] software. Schemes were built with Pymol version 2.2.2 software and edited with Adobe Photoshop version 22.5.0.

### 5.13. Access Numbers

The sequences of each of the toxins were deposited in the UniProt knowledgebase portal under the following access numbers: C0HM15 for Chui2, C0HM16 for Chui3, C0HM17 for Chui4, and C0HM18 for Chui5.

## Figures and Tables

**Figure 1 toxins-14-00369-f001:**
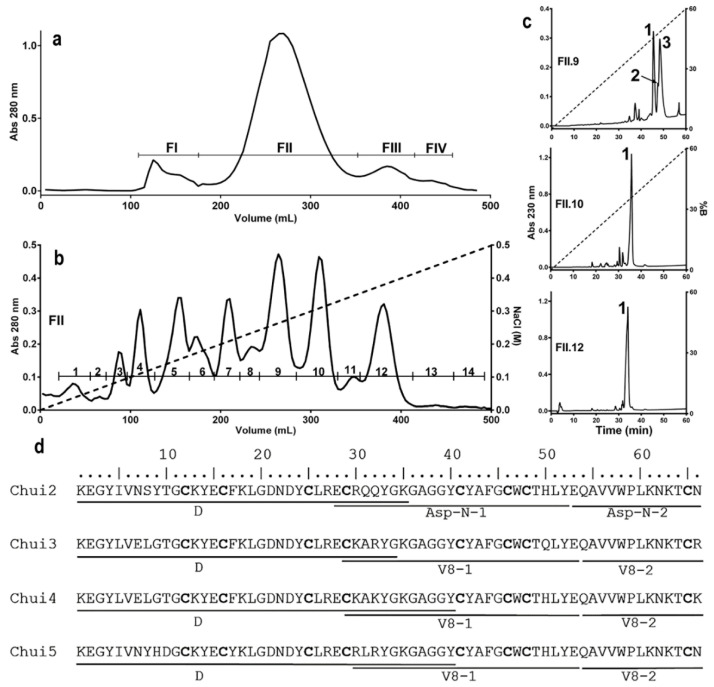
Characterization of the toxic components of *C. huichol* venom. A sample corresponding to 107 mg of protein from fresh venom of the *C. huichol* scorpion was processed to purify the toxic components. (**a**) Molecular exclusion chromatogram of the fresh venom. (**b**) Ion exchange chromatogram of the FII fraction showing 14 sub-fractions. (**c**) Reversed-phase chromatograms of subfractions FII.9 (Chui1, Chui2 and Chui3), FII.10 (Chui4), and FII.12 (Chui5). (**d**) Alignment of the sequences of Chui2, Chui3, Chui4, and Chui5 toxins showing the direct sequencing (D) of the first 34–40 amino acids and of the fragments generated by the Asp-N and V8 proteases. Cysteine residues are indicated in bold.

**Figure 2 toxins-14-00369-f002:**
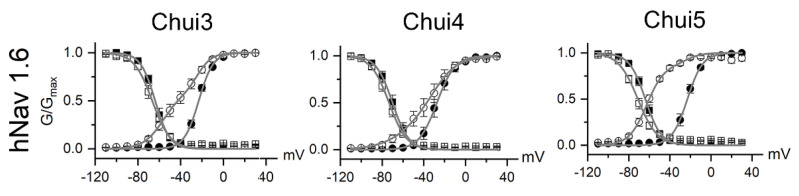
Effect of the *C. huichol* toxins on sodium channels. The figure shows the voltage dependence of activation (circles) and inactivation (squares) process of the hNav 1.6 channel under control condition (closed symbols) and for toxins at 200 nM (open symbols). Data were fitted by means of Boltzmann equation (grey line). For hNav 1.6 with Chui3, Chui4, and Chui5 toxins, data were fitted with double Boltzmann equations, where one of them has the same parameter of the corresponding control condition. The activation and inactivation fitting parameters, as well as the fractional residual current values are summarized in Appendix A.

**Figure 3 toxins-14-00369-f003:**
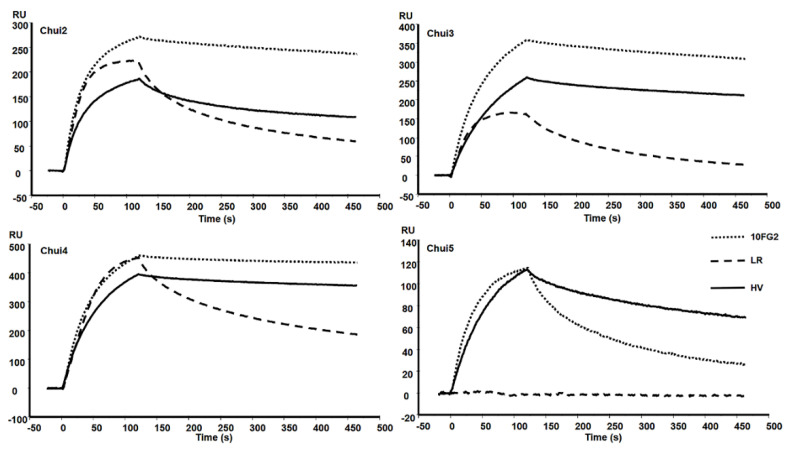
Sensorgrams of the interaction of the indicated scFvs with the toxins of *C. huichol.* The toxins were bound to a CM5 Chip and the interaction of the scFvs 10FG2 and HV at a concentration of 100 nM and of LR at 25 nM was evaluated. Sensorgrams were obtained at the standard temperature of 25 °C.

**Figure 4 toxins-14-00369-f004:**
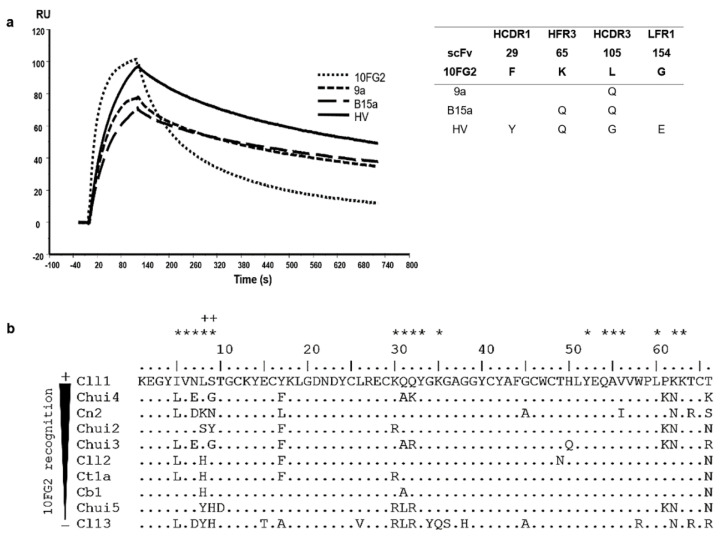
Affinity maturation cycles of scFv 10FG2 against Chui5 toxin. (**a**) Sensorgrams showing the interaction of scFv 10FG2 and the three variants obtained through corresponding three maturation processes at a concentration of 100 nM against Chui5 toxin. The adjacent table describes the mutations acquired by each of the variants, as well as the positions in which they are found. (**b**) Scorpion toxins aligned according to the level of recognition they show against scFv 10FG2. The analysis was performed evaluating the amino acid changes that the toxins present in the different reported interaction sites (asterisks) with the scFv 10FG2. In this way, positions 8 and 9 in the toxins (as well as amino acids K65 and L105 in the scFv) whose changes affect the interaction (indicated by the plus sign) were identified.

**Figure 5 toxins-14-00369-f005:**
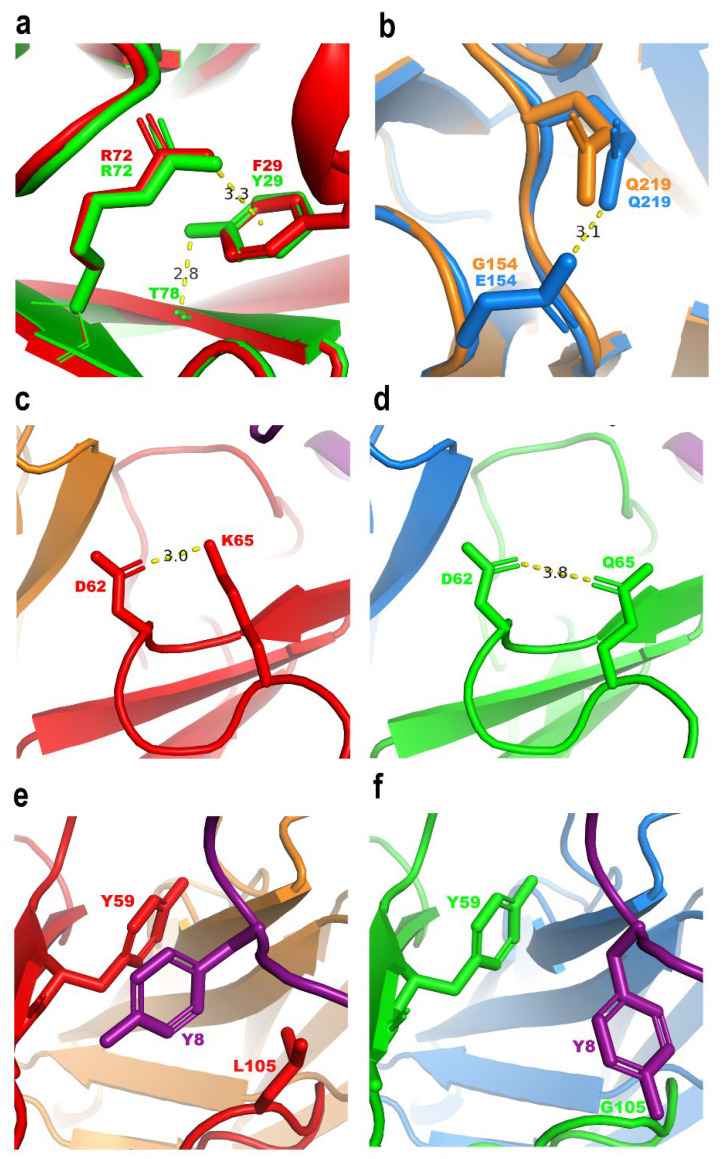
Structural comparison of the in silico models of binary Chui5-10FG2 and Chui5-HV complexes. (**a**) F29 of scFv 10FG2 forms a cation-π interaction with R72 of the same scFv, which is conserved in HV containing residue Y29. In addition, the HV Y29 forms a new hydrogen bond with the main chain of T78. (**b**) The E154 mutation in HV presumably allowed the formation of a new hydrogen bond with Q219, which could not be formed with G154 of 10FG2. (**c**,**d**) K65 of 10FG2 forms a salt bridge with D62. The salt bridge is lost by exchanging it for Q65 in HV, but a weak hydrogen bond with D62 is gained. (**e**) L105 in 10FG2 promotes a conformational change of Y8 in Chui5, showing a steric hindrance with Y59 of 10FG2. (**f**) By mutating position 105 to a small amino acid such as glycine, Y8 is now in a looser condition and without the steric hindrance with Y59. The red and orange colors represent the variable domains of the heavy and light chains of the scFv 10FG2, respectively; the green and blue colors represent the variable domains of the heavy and light chains of the scFv HV, respectively; the purple color corresponds to the Chui5 toxin. Distances are indicated in angstroms (Å).

**Table 1 toxins-14-00369-t001:** *C. huichol* venom neutralization assay with scFvs LR and 10FG2.

Venom	μg/20 g of Mouse	LD_50_	Molar Ratio	Survivors (Alive/Total)	
				Control	Mix-Type Assay
*C. huichol*	15.4	1	1:10:10	1/4 ***	3/4 **

Signs of intoxication are indicated by ** moderate (tail wagging, abdominal twitching); *** strong (sialorrhea, difficult breathing, paralysis of hind legs, death). Control group was injected with 1 LD_50_ of venom. The molar ratio shown refers to venom: scFv LR: scFv 10FG2. For this assay, a proportion of 10 % of the toxins within the venom was assumed [16,18,20].

**Table 2 toxins-14-00369-t002:** Characteristics of *C. huichol* toxins.

Sub-Fraction	Name	Toxicity	Experimental MW (Da)	Theoretical MW (Da)	Relative Abundance (%)
FII.9-1	Chui1	No	7405.8	ND	5.9
FII.9-2	Chui2	Yes	7633.5	7633.7	0.3
FII.9-3	Chui3/Cn3	Yes	7543.9	7544.7	1.8
FII.10-1	Chui4	Yes	7496.7	7497.6	6.2
FII.12-1	Chui5	Yes	7725.6	7726.8	5.7

Toxicity of all fractions was evaluated at a dose of 1.7 μg/20 g of mice. ND: not determined; because the amino acid sequence was not obtained. Chui3 toxin has the same sequence as the Cn3 toxin of *C. noxius*.

**Table 3 toxins-14-00369-t003:** Kinetic parameters of the interaction of scFvs 10FG2 and HV with *C. huichol* toxins.

scFv	Toxin	k_on_ (×10^5^ M^−1 ^s^−1^)	k_off_ (×10^−4^ s^−1^)	T_R_ (min)	K_D_ (nM)
10FG2	Chui2	2.8	2.8	60.4	1.1
	Chui3	1.8	3.1	53.1	1.8
	Chui4	2.1	0.7	241.5	0.3
	Chui5	2.6	34.1	4.9	12.2
HV	Chui2	1.6	4.9	34.2	3.1
	Chui3	0.9	3.4	48.8	3.8
	Chui4	1.8	1.6	104.8	0.9
	Chui5	1.6	10.8	15.5	6.9

**Table 4 toxins-14-00369-t004:** Neutralization of *C. huichol* toxins with scFv 10FG2.

Toxin	1 LD_100_ (μg/20 g of Mouse)	Molar Ratio	Survivors (Alive/Total)	
			Control	Mix-Type Assay
Chui3	2	1:10	0/6 ***	6/6
Chui4	2.5	1:10	0/6 ***	6/6
Chui5	2	1:10	0/6 ***	2/6 ***

Control and experimental groups were injected with 1 LD_100_ of the different toxins. Signs of intoxication are indicated by *** strong (sialorrhea, difficult breathing, paralysis of hind legs, death).

**Table 5 toxins-14-00369-t005:** Determination of the kinetic parameters of Chui5-scFvs interaction.

scFv	k_on_ (×10^5^ M^−1 ^s^−1^)	k_off_ (×10^−4^ s^−1^)	T_R_ (min)	K_D_ (nM)	Yield (mg/L)
10FG2	2.6	34.1	4.9	12.2	3.1
9a	1.5	11.9	14.0	8.2	2.3
B15a	1.3	9.7	17.1	7.3	1.5
HV	1.6	10.8	15.5	6.9	1.7

**Table 6 toxins-14-00369-t006:** Neutralization assays of the matured scFvs against Chui5 toxin.

scFv	LD_50_	μg/20 g of Mouse	Molar Ratio	Appearance of Signs (min)	Survivors (Alive/Total)	
					Control	Mix-Type Assay
-	2	2.2	-	8	0/6 ***	-
10FG2	2	2.2	1:10	18	0/6 ***	1/6 ***
9a	2	2.2	1:10	99	0/4 ***	3/5 **
B15a	2	2.2	1:10	125	0/6 ***	2/6 **
HV	2	2.2	1:10	138	0/4 ***	3/4 *
HV	1	1.1	1:20	NS	3/6 ***	6/6

Control groups with a 1:10 molar ratio were injected with 2 LD_50_ of toxin. Control group with molar ratio 1:20 was injected with 1 LD_50_ of toxin. The molar ratio represents toxin: scFv. NS: no signs of envenoming. Signs of intoxication are indicated by * minimal (ruffled hair, itching); ** moderate (tail wagging, abdominal twitching); *** strong (sialorrhea, difficult breathing, paralysis of hind legs, death).

**Table 7 toxins-14-00369-t007:** Neutralization of *C. huichol* venom with scFv HV.

Assay	Group	LD_50_	Molar Ratio	Survivors (Alive/Total)
	Control	2	-	0/6 ***
	HV	1	1:10	6/6 *
Mix-type		2	1:5	4/6 *
		3	1:3.3	2/6 **
	10FG2 + HV	1	1:10:10	6/6
		5	1:5:5	6/6 **
Rescue	Control	3	-	0/6 ***
	HV	3	1:10	6/6 **

Molar ratio represents venom: scFv HV or venom: scFv 10FG2: scFv HV. The molar ratio for this experiment is given considering that the proportion of the toxins within the venom is 14%. Signs of intoxication are indicated by * minimal (ruffled hair, itching); ** moderate (tail wagging, abdominal twitching); *** strong (sialorrhea, difficult breathing, paralysis of hind legs, death).

**Table 8 toxins-14-00369-t008:** Thermodynamic parameters of Chui5-scFv complexes.

scFv	ΔG (kcal·mol^−1^)	ΔH (kcal·mol^−1^)	ΔS (cal·mol^−1^·K^−1^)
10FG2	−10.8	−2.6	27.3
B15a	−11.1	−2.6	28.6
HV	−11.1	−1.6	31.9

## Data Availability

All data are included within the article and Appendix A. Additional data supporting this study are available from the corresponding author upon reasonable request.

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
