# Peer review of "Characterization of Four Medically Important Toxins from Centruroides huichol Scorpion Venom and Its Neutralization by a Single Recombinant Antibody Fragment"

_toxins, 2022, doi:10.3390/toxins14060369_

Round 1

Reviewer 1 Report

The manuscripts titled “Characterization of Four Medically Important Toxins from Centruroides huichol Scorpion Venom and its Neutralization by a Single Recombinant Antibody Fragment” reported the development of a useful recombinant antibody to neutralize the scorpion toxin by phage display screening. It is an excellent and unquestionably a good paper. I could not find a serious problem in the manuscript.

Author Response

Dear Reviewer, we really appreciate your stimulating comments that value the effort made in this work.

Reviewer 2 Report

The work “Characterization of Four Medically Important Toxins from Centruroides huichol Scorpion Venom and its Neutralization by a Single Recombinant Antibody Fragment” is devoted to determining molecular basis of toxicity of C. huichoi venom and development of antivenom, based on recombinantly produced single-chain antibody variable fragments. The venom was shown to contain four toxins: Chui2, 3, 4, 5. They are rather homologous, but only three of them are neutralized by a single scFv 10FG2. The work is aimed at modifying this scFv. The resulting mutated scFv HV protected 100 % of the mice injected with 3 lethal doses LD50 of the venom. The work is very rich in the experimental data. However, the manner of the presentation of the data makes the understanding of the work difficult.

1. The are some clumsy expressions across the manuscript (few examples: line 24 “in order of two hundred seventy thousand”, line 289 “the sum of two Boltzmann equations”). There are lengthy explanations, e.g. line 877 (where m is the slope and b is the y-intercept, so that). These are well-known facts and the corresponding sentence can be made shorter.

2. References to unpublished data are present: line 333, 699, 782. The volume of the Supplementary file is small. Why not to place some unpublished data to it? E.g. amino acid sequences of scFv HV, 10FG2 ? Figure S2 is not enough. Could it be redrawn with adding this info? This information is necessary to understand in silico modelling of the interaction of Chui5 toxins with scFv. The work refers to PDB model of the complex of another toxin with scFv LR and RU1. Is amino acid difference between scFv RU1 and scFv HV is small enough and their spatial models are similar? This should be specified in the text of the manuscript.

3. As far as Figures is concerned. Fig. 2 contains many panels which are looking very similar. Only typical curves are worth to be shown, taking into account that he fitting parameters are given in the Table S1. Fig. 5 displays trivial lines. Table 8 is enough to be mentioned in the main part. Fig. 5 deserves to be replaced in the Supplementary part. The order of the panels in Figure 7 seems strange. One expects the panels a,b,c in the first row. Then d,e,f in the second one. And, finally, g,h in the last row. In fact, the panels appear in the following order: a, c, d, b, e, f, g, h. Also, panels e,g are low informative and can be eliminated. The resulting Figure would me more compact.

4. The work wins substantially if all abbreviations are summarized in one place, e.g in page 1 of the manuscript.

Author Response

Dear Reviewer, thank you very much for your corrections.
Please see the attached file where you will find all the responses.
